# Identification of quiescent FOXC2+ spermatogonial stem cells in adult mammals

Zhipeng Wang[1][†], Cheng Jin[1][†], Pengyu Li[1], Yiran Li[1], Jielin Tang[1], Zhixin Yu[1], Tao Jiao[1], Jinhuan Ou[1], Han Wang[1], Dingfeng Zou[1], Mengzhen Li[1], Xinyu Mang[1], Jun Liu[1], Yan Lu[1], Kai Li[1], Ning Zhang[2], Jia Yu[1], Shiying Miao[1], Linfang Wang[1], Wei Song[1]*

[1]Department of Biochemistry and Molecular Biology, State Key Laboratory of Common Mechanism Research for Major Diseases, Institute of Basic Medical Sciences, Chinese Academy of Medical Sciences and Peking Union Medical College, Beijing, China; [2]Medical Research Council Protein Phosphorylation and Ubiquitylation Unit (MRC-PPU), School of Life Sciences, University of Dundee, Dundee, United Kingdom

*For correspondence:
songwei@ibms.pumc.edu.cn

[†] These authors contributed equally: Zhipeng Wang, Cheng Jin.

Competing interest: The authors declare that no competing interests exist.

**Abstract** In adult mammals, spermatogenesis embodies the complex developmental process from spermatogonial stem cells (SSCs) to spermatozoa. At the top of this developmental hierarchy lie a series of SSC subpopulations. Their individual identities as well as the relationships with each other, however, remain largely elusive. Using single-cell analysis and lineage tracing, we discovered both in mice and humans the quiescent adult SSC subpopulation marked specifically by forkhead box protein C2 (FOXC2). All spermatogenic progenies can be derived from FOXC2+ SSCs and the ablation of FOXC2+ SSCs led to the depletion of the undifferentiated spermatogonia pool. During germline regeneration, FOXC2+ SSCs were activated and able to completely restore the process. Germ cell-specific *Foxc2* knockout resulted in an accelerated exhaustion of SSCs and eventually led to male infertility. Furthermore, FOXC2 prompts the expressions of negative regulators of cell cycle thereby ensures the SSCs reside in quiescence. Thus, this work proposes that the quiescent FOXC2+ SSCs are essential for maintaining the homeostasis and regeneration of spermatogenesis in adult mammals.

## eLife assessment

This **important** study reports that Foxc2+ cells in the testis represent the quiescent spermatogonial stem cells (SSCs). The data supporting this claim are **solid**. The finding is of great significance to reproductive and stem-cell biology as male fertility depends on the fine balance between self-renewal and differentiation activities of the male germline stem cells, i.e., SSCs.

## Introduction

Through spermatogenesis, spermatozoa are generated from spermatogenic cells that are originated from spermatogonial stem cells (SSCs). It is critical for this process to be continuous and successful that SSCs are maintained in a homeostatic balance between self-renewal and differentiation (*Sharma et al., 2019b*). The SSCs, which belong to a subgroup of undifferentiated spermatogonia (uSPG), exhibit significant heterogeneity and dynamic characteristics. In recent years, great insights into SSC behaviors and regulations have been provided by a body of pioneer works, especially with recent

advances in single-cell gene expression profiling, highlighting great heterogeneity of SSC and focusing on characterizing the nature of SSC states. Within the population of uSPG, a number of genes relatively higher expressed in primitive subfractions have been identified and well investigated, that is, *Gfra1*, *ID4*, *Ret*, *Eomes*, *Pax7*, *Nanos2*, *Shisa6*, *T*, *Pdx1*, *Lhx1*, *Egr2*, and *Plvap* (*Aloisio et al., 2014*; *Guo et al., 2004*; *Hara et al., 2014*; *Helsel et al., 2017*; *Jijiwa et al., 2008*; *La et al., 2018*; *Nakagawa et al., 2021*; *Oatley et al., 2007*; *Sada et al., 2009*; *Sharma et al., 2019a*; *Tokue et al., 2017*). Particularly, *Gfra1*, *ID4*, *Eomes*, *Pax7*, *Nanos2*, and *Plvap* are further validated through lineage tracing experiment, which is considered to be a reliable method to study the development of stem cells. However, some essential and primitive subpopulations remain undiscovered, and the identification of which is of great significance for elucidating the developmental process of SSC renewal and its behavior in testis.

Adult stem cells, as the undifferentiated primitive cells that can be found in nearly all types of tissues in mammals, are characteristic for a unique quiescent status reflected by both reversible cell cycle arrest and specific metabolic alterations (*van Velthoven and Rando, 2019*). The role of the quiescent stem cells is to avoid premature exhaustion and to allow the long-term maintenance of a functional stem cell pool (*Cheung and Rando, 2013*). The adult SSCs appear to share this characteristic, as revealed in recent single-cell RNA-sequencing (scRNA-seq) analysis in humans and mice, being largely non-proliferative while capable of reciprocating between the quiescent and activated status during homeostasis and regeneration (*Guo et al., 2018*; *Hermann et al., 2018*; *Suzuki et al., 2021*; *Tan and Wilkinson, 2019*; *Wang et al., 2018*). This notion has also been significantly supported by the discoveries of important regulators such as USF1 (*Faisal et al., 2019*) and DNMT3L (*Liao et al., 2014*) as well as the pathways including the PI3K/MAPK and mTORC1 signalings (*Suzuki et al., 2021*). In addition, it is generally believed that cells in a quiescent state are supposed to be more resilient to genotoxic insults, thus theoretically the quiescent SSCs can restore spermatogenesis upon such disturbance. However, by large, much remain unknown as to the characteristics and regulation of the quiescence state of adult SSCs. Thus, more important insights can be obtained through searching the quiescent SSC population and defining the essential characteristics of this population as the foundation of successful spermatogenesis.

Forkhead box protein C2 (FOXC2), a member of the forkhead/winged helix transcription factor family, plays essential roles in the development of various tissues in mice (*Bahuau et al., 2002*; *Kume et al., 2001*; *Motojima et al., 2016*; *Sasman et al., 2012*). Recently, FOXC2 has been found to be expressed in $A_s$ and $A_{pr}$ spermatogonia of mouse testes and is required for SSC maintenance (*Wei et al., 2018*). However, the role of FOXC2 in the development of adult SSCs in vivo has yet to be explored. In this study, we identified a subpopulation of adult SSCs specifically expressed by FOXC2. In adult mice, FOXC2[+] SSCs gave rise to all spermatogenic progenitor cells that can complete the full spermatogenesis. Upon the loss of FOXC2[+] SSCs, the uSPG pool was exhausted, eventually leading to defective spermatogenesis. Specifically, FOXC2 is required for maintaining SSC quiescence by promoting the expression of negative regulators of cell cycle. Moreover, the FOXC2[+] population endured the chemical insult with busulfan and effectively restored spermatogenesis. Overall, we propose that FOXC2[+] represents the quiescent state of the SSCs in adult mammals that is crucial for the homeostasis and regeneration of SSCs.

## Results

### Identification of FOXC2[+] SSCs from the adult uSPGs

We performed scRNA-seq (10x genomics) of the uSPG from adult mice testes marked by THY1, a widely recognized surface marker for uSPG with self-renewing and transplantable state (*Hammoud et al., 2014*; *Kubota et al., 2004*), to dissect the heterogeneity and developmental trajectory (*Figure 1A*, *Figure 1—figure supplement 1A, B*). Among five distinct clusters identified, Cluster1 was characterized by the high expression of stemness markers whereas other clusters were featured by progenitor or differentiating spermatogonia (dSPG) markers (*Figure 1B*, *Figure 1—figure supplement 1C, D*). Primarily mapped to the extreme early point of the developmental trajectory, Cluster1 cells appeared quiescent and likely represented the primitive state of uSPG populations (*Figure 1B*, *Figure 1—figure supplement 1E–G*). The top 10 differentially expressed genes (DEGs) associated with Cluster1 are featured by SSC markers such as *Mcam* (*Kanatsu-Shinohara et al., 2012*), *Gfra1* (*Hara et al., 2014*),

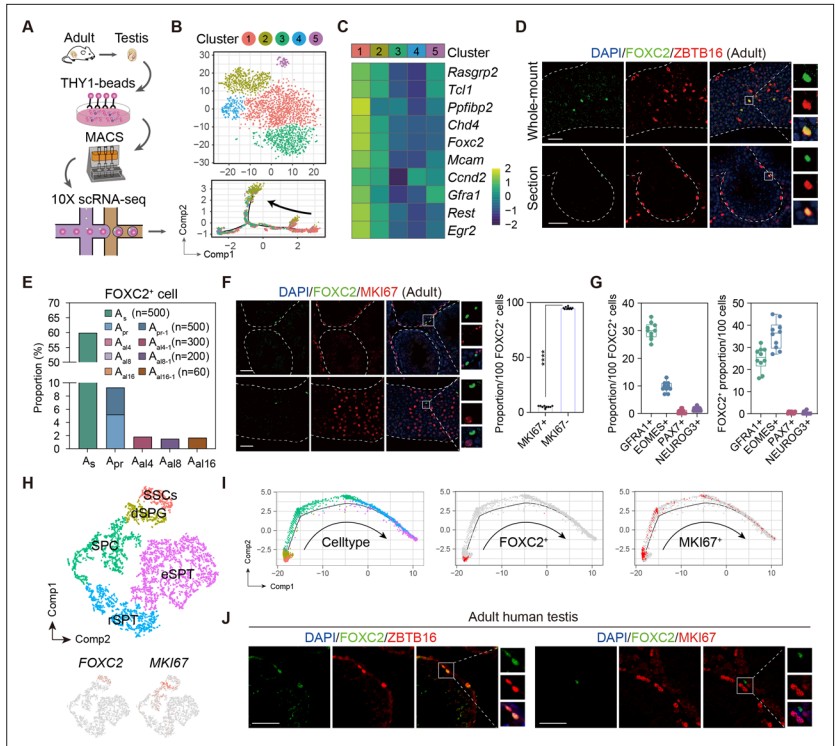

**Figure 1.** Identification of the FOXC2+ spermatogonial stem cells (SSCs) in adult mouse and human testis. (**A**) Schematic illustration of the single-cell analysis workflow. (**B**) t-SNE plot and developmental trajectory of all undifferentiated spermatogonia (uSPG), colored by cluster. (**C**) Heatmap of the top 10 differentially expressed genes (DEGs) in Cluster1. (**D**) Immunostaining for ZBTB16 (red), FOXC2 (green), and DAPI (blue) in testicular paraffin sections from wild-type adult C57 mice. Scale bar, 50 µm; C57, C57BL/6J. (**E**) The proportion of FOXC2+ cells in different uSPG subtypes. (**F**) Immunostainings for MKI67 (red), FOXC2 (green), and DAPI (blue) in adult mice testis and the proportion of MKI67+ cells in FOXC2+ population (n=10). Scale bar, 50 µm; values, mean ± s.e.m.; p-values were obtained using two-tailed t-tests (****p-value <0.0001). (**G**) The co-expression proportion between the FOXC2 and differential known SSCs makers (n=10). (**H**) t-SNE plot of germ cells in adult human testis (GSE112013), colored by germ cell type. Feature plot showing the expression patterns of FOXC2 and MKI67 in human germ cells. (**I**) The developmental trajectory of the human germ cells, colored by germ cell type, FOXC2 expression cells (red), or MKI67 expression cells (red). (**J**) Immunostaining for ZBTB16/MKI67 (red), FOXC2 (green), and DAPI (blue) in testicular paraffin sections from adult humans.

The online version of this article includes the following source data and figure supplement(s) for figure 1:

**Source data 1.** Excel spreadsheet with the list of the top 30 differentially expressed genes of different clusters.

**Figure supplement 1.** Validation and characterization of the magnetic-activated cell sorting (MACS)-sorted THY1+ undifferentiated spermatogonia (uSPG) from wild-type adult C57 mice.

**Figure supplement 2.** Expression of top 10 differentially expressed genes (DEGs) of Cluster1 in *Figure 1B* and classic spermatogonial stem cell (SSC) and SPG markers in adult human germ cells.

---

*Tcl1* and *Egr2* (*Hermann et al., 2018*; *La et al., 2018*; *Figure 1C*, *Figure 1—figure supplement 2A*, *Figure 1—source data 1*) in addition to six others expressed in different stages of germ cells and/or somatic cells, in which only FOXC2 was exclusively localized in the nucleus of a subgroup of ZBTB16+ uSPG in mice (*Buaas et al., 2004*; *Costoya et al., 2004*; *Figure 1D*, *Figure 1—figure supplement 2B*). More specifically, FOXC2 displayed differential expressions among various subtypes of uSPG, being more specific in $A_s$ (59.9%) than other subtypes including $A_{pr}$ (5.2%), $A_{pr-1}$ (4.1%), $A_{al4-1}$ (1.83%), $A_{al8-1}$ (1.5%), and $A_{al16-1}$ (1.67%) (*Figure 1E*). There was only a small fraction (5.1%) active in proliferation as indicated by MKI67 (*Figure 1F*), suggesting that FOXC2+ cells are primarily quiescent. Additionally, when examining the SSC markers validated previously by lineage tracing (*La and Hobbs, 2019*), we found that FOXC2 displays a higher level of co-localization with GFRA1 and EOMES than PAX7 and

NEUROG3 (*Nakagawa et al., 2007*), indicating the FOXC2$^+$ cells contain but differ from the known SSC subsets (*Figure 1G*).

We next analyzed the expression of FOXC2 in adult human testis using the published scRNA-seq dataset (GSE112013) (*Guo et al., 2018*). As expected, FOXC2 was also specifically expressed in the human SSCs, most of which were MKI67$^-$ (*Figure 1H*, *Figure 1—figure supplement 2C*). Pseudotime analysis showed that the FOXC2$^+$ cells located at the start of the developmental trajectory with a proportion of about 90% that were MKI67$^-$ (*Figure 1I*). Immunofluorescence staining confirmed that FOXC2$^+$ cells were a subset of ZBTB16$^+$ spermatogonia in adult human testis, and most of them were MKI67$^-$ (*Figure 1J*), representing a quiescent subpopulation of SSCs in human testis (*Clermont, 1966a*; *Clermont, 1966b*; *Clermont, 1969*; *Ehmcke and Schlatt, 2006*; *Ehmcke et al., 2006*). These results suggest that FOXC2 is similarly expressed in the SSCs of adult human and mouse testis and may possess a conserved function.

## FOXC2$^+$ SSCs sufficiently initiate and sustain spermatogenesis

We generated *Foxc2$^{iCreERT2/+}$;Rosa26$^{LSL-T/G/LSL-T/G}$* mice in which FOXC2$^+$ cells were specifically labeled with GFP to enable the progeny tracing after tamoxifen treatment (*Figure 2—figure supplement 1*; *Wang et al., 2015*). Tamoxifen was introduced at 2 month of age, after which the FOXC2-expressing lineage (GFP$^+$) was tracked at d3 (day3), w1 (week1), w2, w4, w6, m4 (month 4), m7, and m12, respectively (*Figure 2A*). At d3, the tracked cells were both GFP$^+$ and FOXC2$^+$ (*Figure 2B*) and constituted 0.027% of the total testicular cells as indicated by the fluorescence-activated cell sorting (FACS) analysis (*Figure 2C*). GFP$^+$ cells and THY1$^+$ cells sorted by FACS were then transplanted into testes of recipient mice pre-treated with busulfan, respectively. Two months after transplantation, FOXC2$^+$ cells generated 3.6 times greater number of colonies than the THY1$^+$ control (*Figure 2D and E*), indicating that the FOXC2$^+$ cells possess higher stemness as convinced by stronger transplantable viability.

At w1, all GFP$^+$ cells were identified as uSPGs, encompassing A$_s$, A$_{pr}$, and A$_{al-4}$ (*Figure 2F$_a$*). Specifically, FOXC2$^+$ A$_s$ gave rise to three types of A$_{pr}$, that is, FOXC2$^+$/FOXC2$^+$, FOXC2$^+$/FOXC2$^-$, and FOXC2$^-$/FOXC2$^-$ (*Figure 2F$_{c1, b, c2, d2}$*), which then either produced FOXC2$^+$ or FOXC2$^-$ A$_s$ through symmetric or asymmetric division (*Figure 2F$_{c3, d1, f1}$*), or developed into A$_{al}$ with no more than one FOXC2$^+$ cell in the chains (*Figure 2F$_{e, f2}$*). These results confirm that FOXC2$^+$ cells are capable of self-renewal to sustain the population as well as replenishing the uSPG pool by producing downstream progenies, thereby serving as SSCs. In the following 2–6 weeks, GFP$^+$ colonies further expanded and produced GFP$^+$ sperms in the epididymis, from which healthy GFP$^+$ offspring were given birth by C57BL/6J female recipients (*Figure 2G*). The GFP$^+$ colonies constituted 83.67%, 90.48%, 96.78%, 98.55%, and 99.31% of the total length of the seminiferous tubules at w6, m2, m4, m7, and m12, respectively (*Figure 2H, I*). All offspring were GFP$^+$ from m4 onward (*Figure 2J*). Additionally, the EOMES$^+$, GFRA1$^+$, and PAX7$^+$ cells were all GFP$^+$ at w2, further confirming these progenies were derived from the FOXC2$^+$ cells (*Figure 2K*). Overall, FOXC2$^+$ SSCs can produce all subtypes of uSPG, thus initiating spermatogenesis in adult mice.

## Specific ablation of the FOXC2$^+$ SSCs results in the depletion of uSPG pool

We then prepared *Foxc2$^{iCreERT2/+}$;Rosa26$^{LSL-DTA/+}$* mice to investigate the physiological requirement of FOXC2$^+$ SSCs in spermatogenesis (*Wang et al., 2015*). FOXC2$^+$ population in 2-month-old mice was specifically ablated with tamoxifen-induced diphtheria toxin (DTA). The testes of these mice were examined at d3, d7, and d14 post tamoxifen induction (*Figure 3A*). Gradual loss of weight in testes coincided with the reduction in the size of testes in all the mice while body weight was maintained (*Figure 3B and C*). Specifically, at d3, there were no detectable FOXC2$^+$ cells in addition to the decrease in the number of GFRA1$^+$, LIN28A$^+$ (*Zheng et al., 2009*), and ZBTB16$^+$ uSPG at the basement membrane of seminiferous tubules; at d14, all GFRA1$^+$, LIN28A$^+$, and ZBTB16$^+$ uSPG disappeared while vacuoles formed at the basement membrane with remaining spermatocytes and spermatids in the seminiferous lumen (*Figure 3D–F*, *Figure 3—figure supplement 1*). Meanwhile, the expression of DDX4 (*Toyooka et al., 2000*) and DAZL (*Li et al., 2019*) as germ cell markers was significantly reduced along with nearly undetectable expression of uSPG markers such as ZBTB16, LIN28A, GFRA1, RET, and NEUROG3 (*Nakagawa et al., 2007*; *Figure 3G*). These results indicate an uSPG exhaustion as the

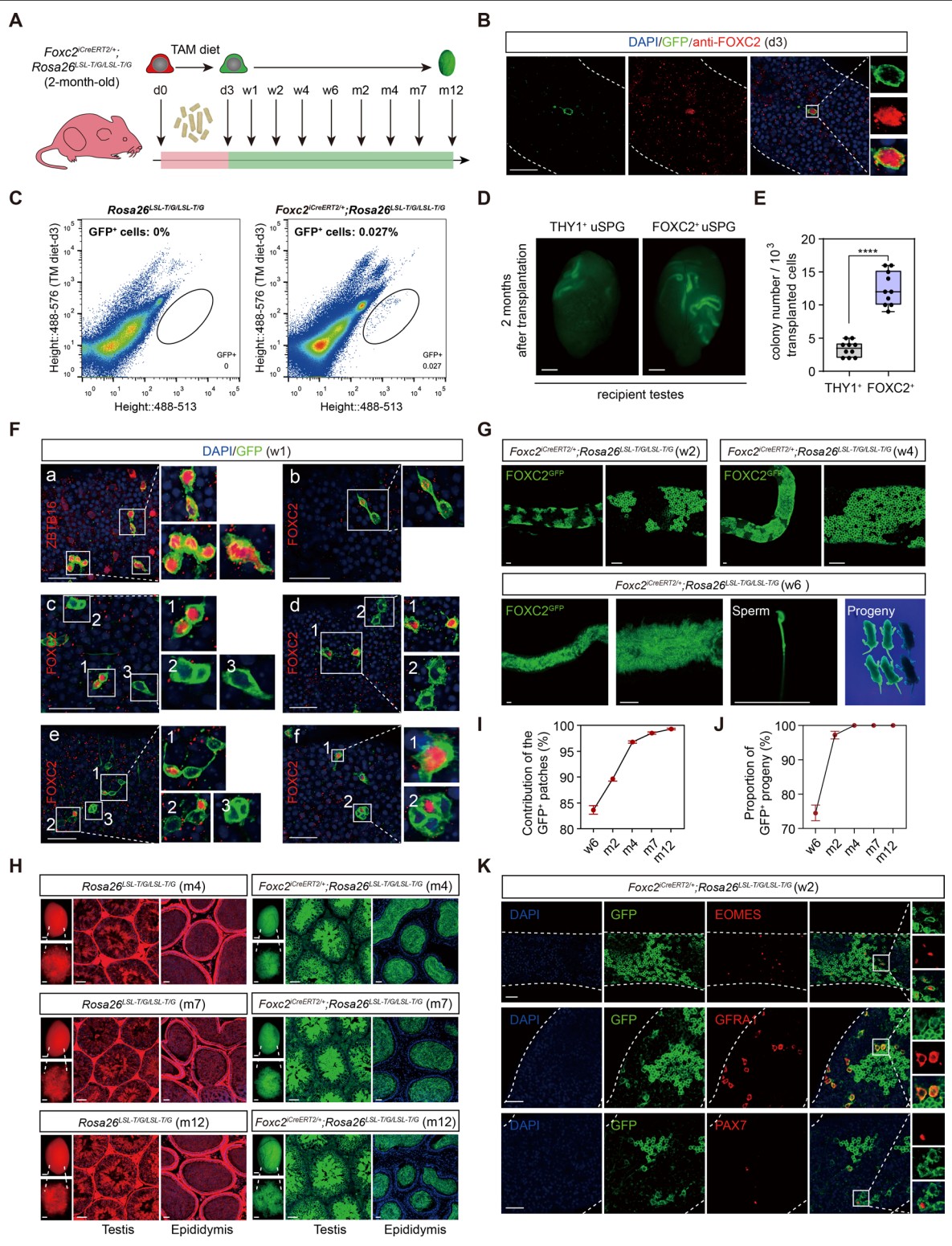

**Figure 2.** Lineage tracing and functional validation of FOXC2⁺ spermatogonial stem cells (SSCs) in *Foxc2^iCreERT2/+;Rosa26^LSL-T/G/LSL-T/G* mice. (**A**) Schematic illustration of the lineage tracing workflow for FOXC2⁺ cells. (**B**) Immunostainings for DAPI (blue) and FOXC2 (red) at day 3 post TAM induction. Scale bar, 50 μm; d, day. (**C**) Fluorescence-activated cell sorting (FACS) analysis of GFP⁺ populations derived from *Rosa26^LSL-T/G/LSL-T/G* or *Foxc2^iCreERT2/+;Rosa26^LSL-T/G/LSL-T/G* mice at day 3 post TAM induction. (**D, E**) The recipient mice testes (**D**) and colony numbers (**E**) 2 months after transplantation (n=10) of the FACS-sorted GFP⁺ cells from the *Foxc2^iCreERT2/+;Rosa26^LSL-T/G/LSL-T/G* mice 3 days after TAM diet and the FACS-sorted THY1⁺ cells from adult mice. Scale bar, 1 mm; values, mean ± s.e.m.; p-values were obtained using two-tailed t-tests (****p-value <0.0001). (**F**) Immunostaining for DAPI (blue), ZBTB16/

*Figure 2 continued on next page*

*Figure 2 continued*

FOXC2 (red), and GFP (green) at week 1 post TAM induction (scale bar, 50 µm). (**G**) Seminiferous tubules of *Foxc2^{iCreERT2/+};Rosa26^{LSL-T/G/LSL-T/G}* mice 2, 4, and 6 weeks post TAM induction. Scale bar, 50 µm. (**H**) Testes (scale bar, 1 mm), seminiferous tubules, and epididymis (scale bar, 50 µm) at months 4, 7, and 12 post TAM induction in *Foxc2^{iCreERT2/+};Rosa26^{LSL-T/G/LSL-T/G}* mice. (**I, J**) The GFP^+ patches (**I**) and progeny (**J**) population dynamics (n=10). Values, mean ± s.e.m. (**K**) Immunostainings for DAPI (blue), EOMES (red), GFRA1 (red), or PAX7 (red) in GFP^+ population at week 2 post TAM induction. Scale bar, 50 µm.

The online version of this article includes the following figure supplement(s) for figure 2:

**Figure supplement 1.** Construction of the *Foxc2^{iCreERt2}* mice.

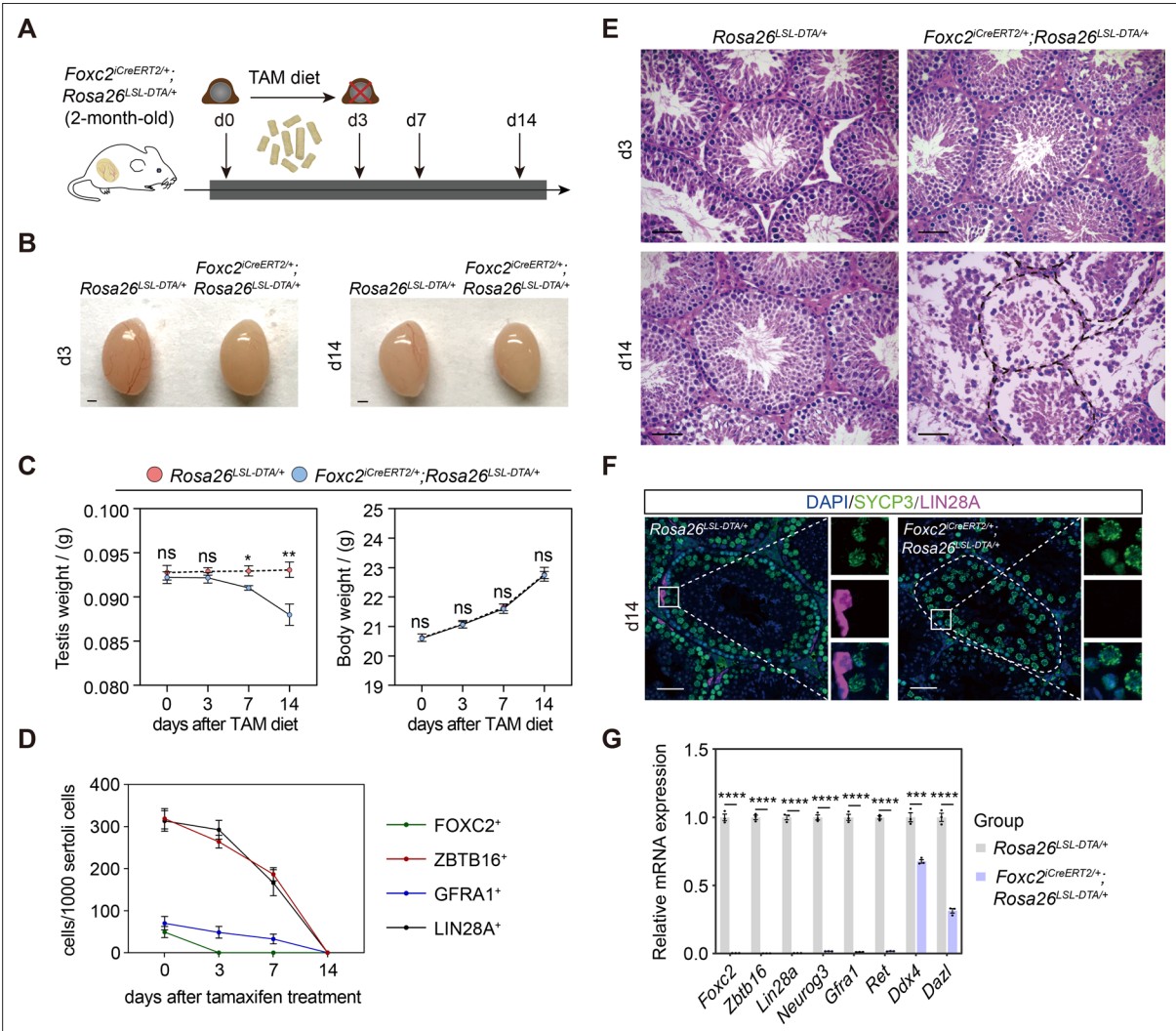

**Figure 3.** Specific ablation of FOXC2^+ spermatogonial stem cells (SSCs) and phenotypic validation in *Foxc2^{iCreERT2/+};Rosa26^{LSL-DTA/+}* mice. (**A**) Schematic illustration of the lineage tracing workflow for FOXC2^+ cells. (**B–D**) Phenotypic validation of the *Rosa26^{LSL-DTA/+}* and *Foxc2^{iCreERT2/+};Rosa26^{LSL-DTA/+}* mice (n=5) for testes size (**B**), testis weight and body weight (**C**), and hematoxylin and eosin (HE) staining of the testes (**D**). Scale bars in (**B**), 1 mm; in (**D**), 50 µm; d, day; values were mean ± s.e.m.; p-values were obtained using two-tailed t-tests (ns >0.05, *p-value <0.05, **p-value <0.01). (**E**) ZBTB16^+, GFRA1^+, LIN28A^+, and FOXC2^+ SPG populations dynamics. Values, mean ± s.e.m. (n=10); p-values were obtained using one-way ANOVA followed by Tukey test (ns >0.05, *p-value <0.05, **p-value <0.01, ****p-value <0.0001). (**F**) Immunostainings for DAPI (blue), SYCP3 (green), and LIN28A (magenta) at day 14 post TAM induction. d, day; scale bar, 50 µm. (**G**) Quantitative RT-PCR analysis of SPG markers expression in the testes of the *Rosa26^{LSL-DTA/+}* and *Foxc2^{iCreERT2/+};Rosa26^{LSL-DTA/+}* mice (n=3). Values, mean ± s.e.m.; p-values were obtained using two-tailed t-tests (***p-value <0.001, ****p-value <0.0001).

The online version of this article includes the following figure supplement(s) for figure 3:

**Figure supplement 1.** Depletion of undifferentiated spermatogonia (uSPG) pool in *Foxc2^{iCreERT2/+};Rosa26^{LSL-DTA/+}* mice 14 days after specific ablation of FOXC2^+ spermatogonial stem cells (SSCs).

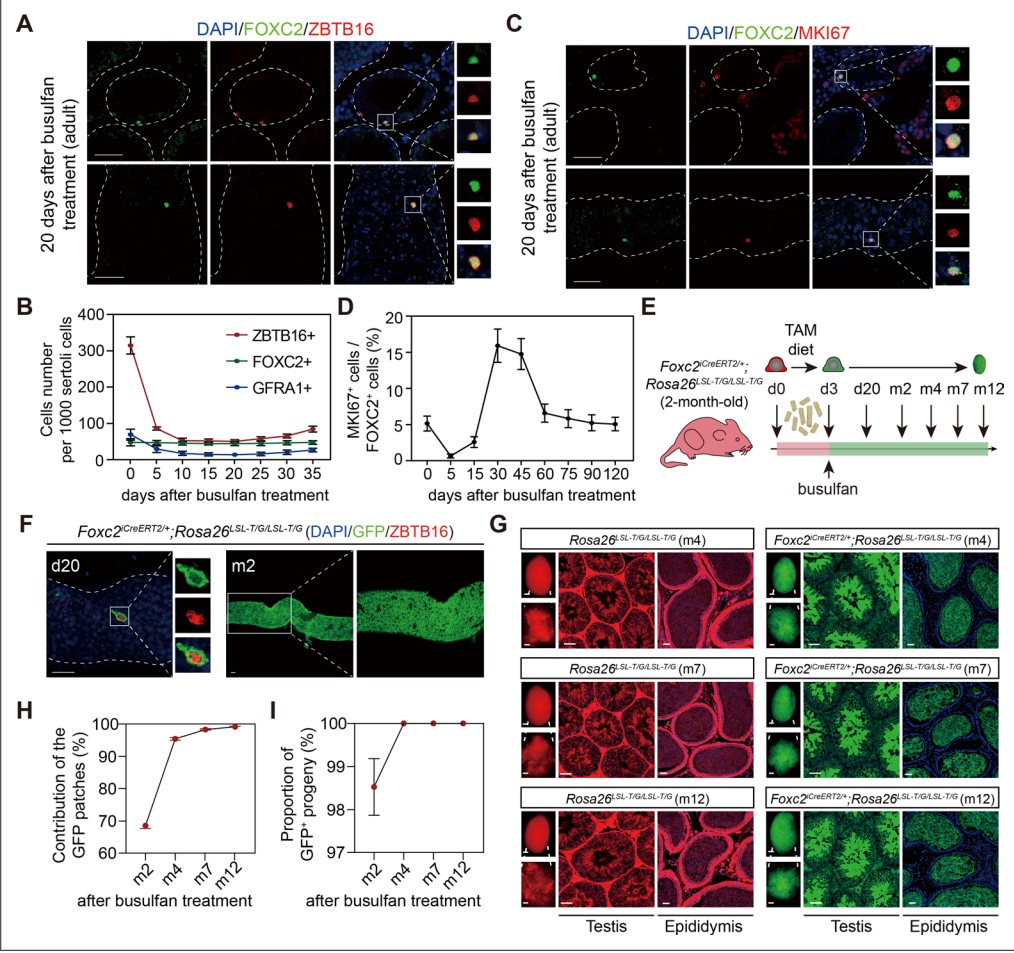

**Figure 4.** FOXC2+ spermatogonial stem cells (SSCs) are critical for germline regeneration. (**A**) Co-immunostaining of FOXC2 (green) with ZBTB16 (red) in seminiferous tubules of the adult testes at day 20 post busulfan treatment. Scale bar, 50 µm. (**B**) ZBTB16+, GFRA1+, and FOXC2+ population dynamics after busulfan treatment (20 mg/kg, n=10). (**C**) Co-immunostaining of FOXC2 (green) with MKI67 (red) in seminiferous tubules of the adult testes at day 20 post busulfan treatment. Scale bar, 50 µm. (**D**) MKI67+FOXC2+ proportions in relation to the whole FOXC2+ population at different time points after busulfan treatment (n=4). (**E**) Schematic illustration for lineage tracing of FOXC2+ cell after busulfan treatment. (**F**) Lineage tracing of the GFP+ cells at day 20 and month 2 after busulfan treatment (scale bar, 50 µm). (**G**) The testes (scale bar, 1 mm), seminiferous tubules, and epididymis (scale bar, 50 µm) at months 4, 7, and 12 post TAM induction and busulfan injection. m, month. (**H, I**) The proportion dynamics of GFP patches (**H**) and GFP+ progenies (**I**). Values, mean ± s.e.m. (n=10). w, week; m, month.

result of the FOXC2+ SSCs ablation, therefore supporting the critical role played by FOXC2+ population in spermatogenesis.

## FOXC2+ SSCs are resistant to busulfan and indispensable for germline regeneration

Next, we examined the regenerative viability of FOXC2+ SSCs. At d20 post busulfan treatment (20 mg/kg), FOXC2+ cells constituted the majority of uSPGs (*Figure 4A*). Following a sharp decrease in cell number in the first 5 days, ZBTB16+ and GFRA1+ cells began to recover from d25 while the number of FOXC2+ cells remained stable (*Figure 4B*), indicating that this population is insensitive to busulfan. We then checked changes in the proportion of MKI67+ cells in FOXC2+ population after busulfan treatment (*Figure 4C and D*). At d30, the MKI67+ proportion rose to 15.92%, indicating a higher level of proliferation, albeit the total cell number stayed static (*Figure 4B and D*), thereby becoming the driving force in restoring spermatogenesis. Up to d120, the MKI67+ proportion had settled gradually back to the pre-treatment level, accompanied by the full recovery of spermatogenesis (*Figure 4D*).

Further details of this process were revealed during lineage tracing (*Figure 4E*). Three days after tamoxifen induction, the 2-month-old *Foxc2$^{iCreERT2/+}$;Rosa26$^{LSL-T/G/LSL-T/G}$* mice were treated with busulfan. Consistent with the results above, at d20, the survived uSPGs were predominantly GFP$^+$ (*Figure 4F*). Over 68.5% of the total length of the seminiferous tubules were GFP$^+$ at m2, and this proportion rose to 95.43%, 98.41%, and 99.27% at m4, m7, and m12, respectively (*Figure 4G and H*), which was comparable to the proportion by tamoxifen induction alone (*Figure 2I*). From m4 onward, nearly all germ cells, spermatids, and their offspring were GFP$^+$ (*Figure 4G, I*). Together, these results confirm that FOXC2$^+$ SSCs are indispensable for germline regeneration that is central to spermatogenesis recovery from interruptions.

## FOXC2 is essential for SSC maintenance in adult mice

We then focused on dissecting the role of FOXC2 in the SSC maintenance using *Foxc2$^{f/-}$;Ddx4$^{Cre/+}$* mice (*Gallardo et al., 2007*; *Figure 5A*). No significant difference was observed in the expressions of various uSPG markers, including ZBTB16 and LIN28A, between *Foxc2$^{f/-}$;Ddx4$^{Cre/+}$* and *Foxc2$^{f/+}$* mice at the age of 1 week (*Figure 5—figure supplement 1A*). However, adult *Foxc2$^{f/-}$;Ddx4$^{Cre/+}$* mice displayed clear testis weight loss without significant body weight loss (*Figure 5B and C*). Moreover, in these mice, we observed severe degeneration of seminiferous tubules, reduced number of spermatids in the epididymis, and decreased size of the uSPG population with age (*Figure 5D–G*) but without apparent signs of apoptosis (*Figure 5—figure supplement 1B*). The 6-month-old *Foxc2$^{f/-}$;Ddx4$^{Cre/+}$* mice were infertile, in which over 95% seminiferous tubules were Sertoli-only with hardly detectable expressions of DAZL, DDX4, LIN28A, and ZBTB16 (*Figure 5D–F and H*). Therefore, FOXC2 is essential for maintaining the SSC homeostasis and normal spermatogenesis in adult mice.

## FOXC2 maintains the quiescent state of SSCs through negatively regulating the cell cycle

We collected THY1$^+$ uSPGs from 4-month-old *Foxc2$^{f/+}$* and *Foxc2$^{f/-}$;Ddx4$^{Cre/+}$* mice and compared their transcriptome signatures revealed from scRNA-seq (*Figure 6A*). The pseudotime analysis identified Cluster1, which represented the FOXC2-expressing SSCs in *Foxc2$^{f/+}$* mice corresponding to the FOXC2-deleting SSCs in the *Foxc2$^{f/-}$;Ddx4$^{Cre/+}$* mice, was specifically assigned to the extremely early stage of the development trajectory in respective samples, which was validated by the expression of corresponding markers (*Figure 6B*, *Figure 6—figure supplement 1A, B*). Aggregated analysis of the overall uSPG populations showed that cells derived from *Foxc2$^{f/-}$;Ddx4$^{Cre/+}$* mice were specifically associated with the late stage of the development trajectory, as opposed to *Foxc2$^{f/+}$* mice where nearly all the cells derived were concentrated at the early stage of development (*Figure 6C*, *Figure 6—figure supplement 1C*). This implies that the loss of FOXC2 prompts the SSCs to progress into a more differentiated stage. Further analysis of the cells in Cluster1 revealed two distinct subclusters, that is, Subclusters0 and Subclusters1 (*Figure 6—figure supplement 2A*). Formed primarily by the Cluster1 cells derived from *Foxc2$^{f/+}$* mice, Subclusters0 was featured by stemness markers, while Subcluster1, representing the majority of Cluster1 cells from *Foxc2$^{f/-}$;Ddx4$^{Cre/+}$* mice, was featured by progenitor markers (*Figure 6—figure supplement 2B, C*). Consistently, pseudotime analysis showed that Cluster1 cells from *Foxc2$^{f/+}$* mice projected a forward stage of the developmental trajectory indicated by stemness markers, whereas Cluster1 cells from *Foxc2$^{f/-}$;Ddx4$^{Cre/+}$* mice were associated with a later stage of the developmental trajectory (*Figure 6D*, *Figure 6—figure supplement 2D, E*). More specifically, less number of cells were found at the starting state1 in Cluster1 from *Foxc2$^{f/-}$;Ddx4$^{Cre/+}$* mice than in *Foxc2$^{f/+}$* mice, with rather more cells in the developmental progression (from state1 to state5), especially at the advanced state5 (*Figure 6E*). Thus, FOXC2 deletion caused defective SSC maintenance and committed the SSCs to a differentiation destiny. Further, there were 932 genes down-regulated in Cluster1 cells derived from *Foxc2$^{f/-}$;Ddx4$^{Cre/+}$* mice in comparison to *Foxc2$^{f/+}$* mice (*Figure 6F*, *Figure 6—source data 1*), which were functionally associated with both stem cell population maintenance and mitotic cell cycle (*Figure 6G*). Consistently, the gene set enrichment analysis (GSEA) revealed a more progressive cell cycle in Cluster1 upon *Foxc2* knockout (*Figure 6H*), confirming the role of FOXC2 in regulating the cell cycle of the SSCs.

We then performed Cleavage Under Targets and Tagmentation (CUT&Tag) sequencing to explore the underlying mechanism (*Kaya-Okur et al., 2020*; *Kaya-Okur et al., 2019*), for which GFP$^+$ SSCs from *Foxc2$^{iCreERT2/+}$;Rosa26$^{LSL-T/G/LSL-T/G}$* mice 3 days after tamoxifen induction, representing the FOXC2$^+$

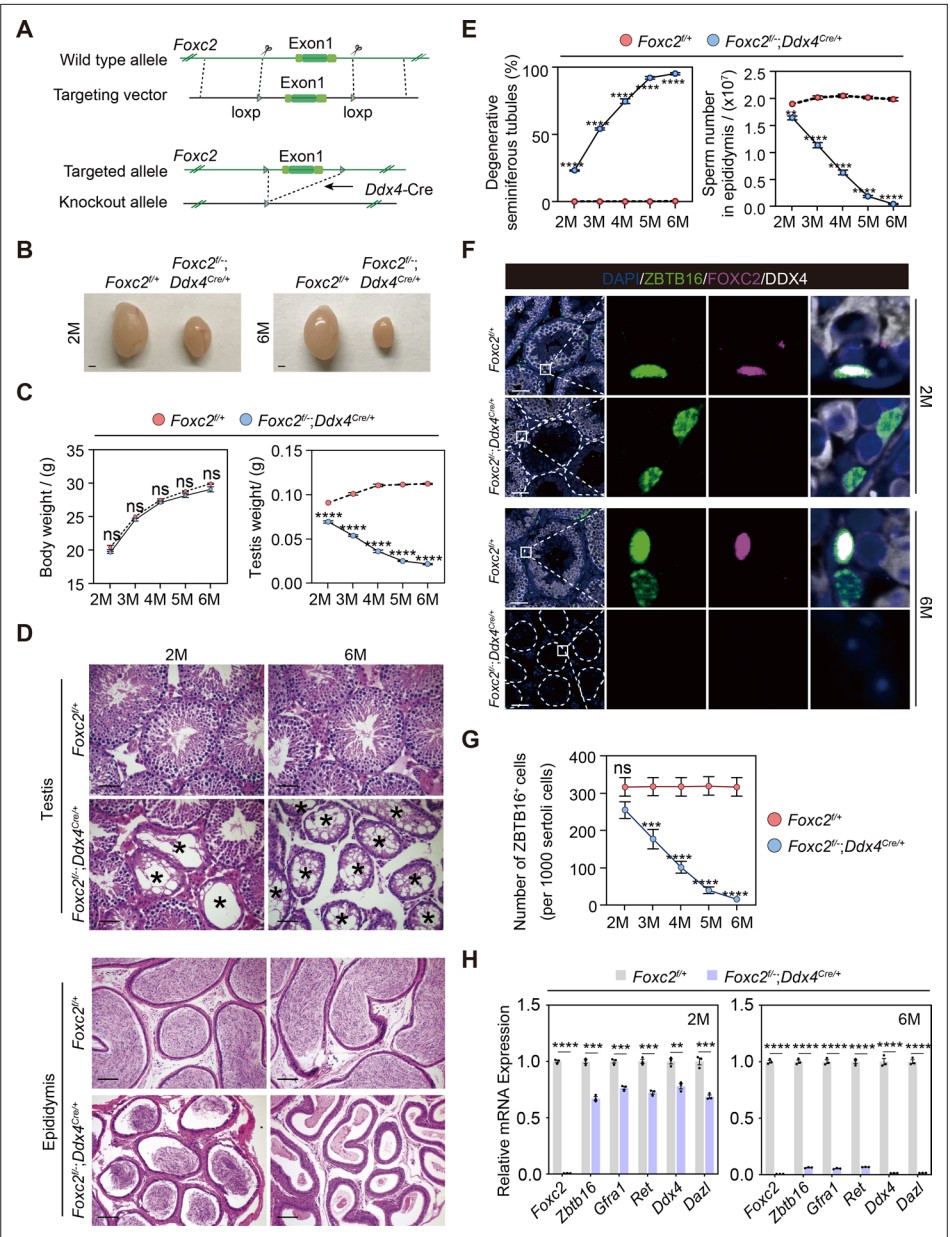

**Figure 5.** Spermatogenesis exhaustion in the adult *Foxc2^f/-^;Ddx4^Cre/+^* mice. (**A**) Construction of the *Foxc2^f/-^;Ddx4^Cre/+^* mice. (**B**) The testes size of the *Foxc2^f/-^;Ddx4^Cre/+^* mice. Scale bar, 1 mm; M, month. (**C**) Body weight and testis weight of the *Foxc2^f/-^;Ddx4^Cre/+^* mice at different age (n=5). M, month; values, mean ± s.e.m.; p-values were obtained using two-tailed t-tests (ns >0.05, ****p-value <0.0001). (**D**) Hematoxylin and eosin (HE) staining of the testis and epididymis. Scale bar, 50 μm; M, month. (**E**) Estimation of degenerative tubules and sperm counts in cauda epididymis of the *Foxc2^f/+^* and *Foxc2^f/-^;Ddx4^Cre/+^* mice with age (n=5). Values, mean ± s.e.m.; p-values were obtained using two-tailed t-tests (**p-value <0.01, ****p-value <0.0001). (**F**) Immunostainings for DAPI (blue), ZBTB16 (green), FOXC2 (magenta), and DDX4 (white) in the seminiferous tubules of the *Foxc2^f/+^* and *Foxc2^f/-^;Ddx4^Cre/+^* mice. Scale bar, 50 μm. (**G**) Estimation of ZBTB16⁺ uSPG number in the *Foxc2^f/+^* and *Foxc2^f/-^;Ddx4^Cre/+^* mice with age (n=5). Values, mean ± s.e.m.; p-values were obtained using two-tailed t-tests (ns >0.05, ***p-value <0.001, ****p-value <0.0001). (**H**) Quantitative RT-PCR analysis of the uSPG and germ cell markers expressed in the testis of the *Foxc2^f/+^* and *Foxc2^f/-^;Ddx4^Cre/+^* mice (n=3). M, month; values, mean ± s.e.m.; p-values were obtained using two-tailed t-tests (**p-value <0.01, ***p-value <0.001, ****p-value <0.0001).

The online version of this article includes the following figure supplement(s) for figure 5:

**Figure supplement 1.** Phenotypic validation of the *Foxc2^f/-^;Ddx4^Cre/+^* mice.

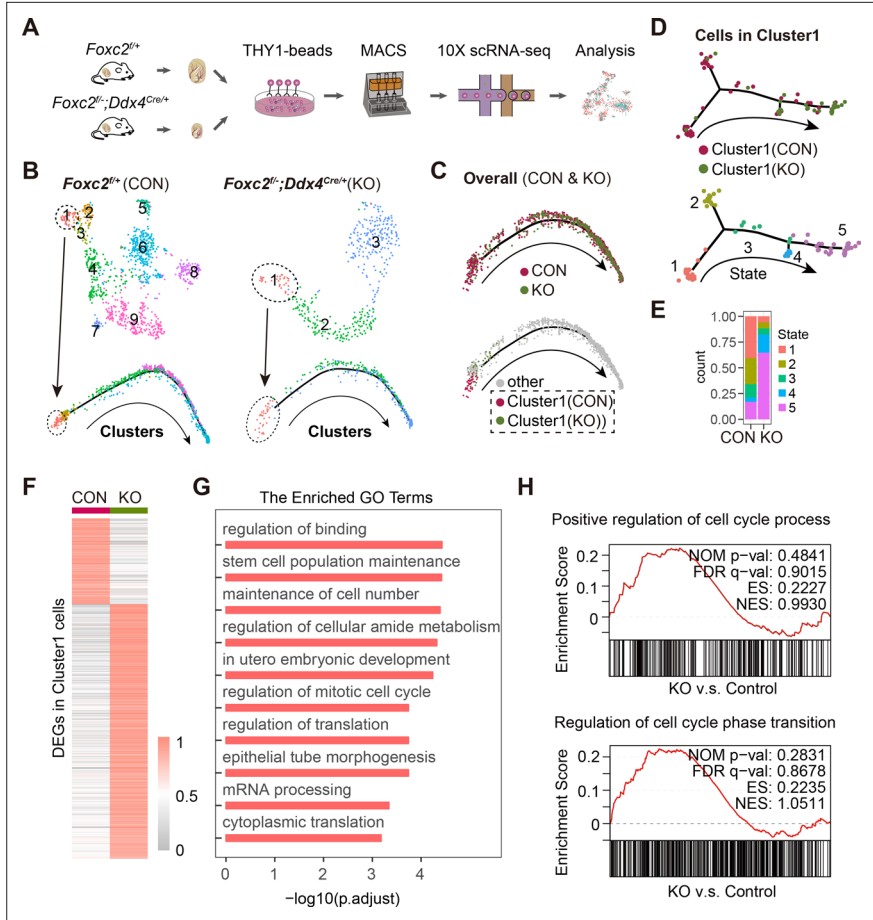

**Figure 6.** Single-cell RNA-sequencing (scRNA-seq) analysis of THY1⁺ undifferentiated spermatogonia (uSPG) in *Foxc2^f/+* and *Foxc2^f/-;Ddx4^Cre/+* mice. (**A**) Schematic illustration of the scRNA-seq workflow. (**B**) t-SNE plot and developmental trajectory of uSPG from *Foxc2^f/+* and *Foxc2^f/-;Ddx4^Cre/+* mice, respectively, colored by cluster. (**C**) Developmental trajectories of uSPG from *Foxc2^f/+* and *Foxc2^f/-;Ddx4^Cre/+* mice, colored by sample or derivation. (**D**) Developmental trajectories of the cells in Cluster1 from *Foxc2^f/+* (CON) and *Foxc2^f/-;Ddx4^Cre/+* (KO) mice, colored by derivation or developmental state. (**E**) The Cluster1 cells proportion of each state in CON and KO mice. (**F**) Heatmap showing the differentially expressed genes (DEGs) in the Cluster1 cells from the *Foxc2^f/-;Ddx4^Cre/+* mice compared with the *Foxc2^f/+* mice. (**G**) Top Gene Ontology (GO) terms enrichment by the down-regulated DEGs in KO mice. (**H**) Gene set enrichment analysis (GSEA) of the Cluster1 cells (*Foxc2^f/-;Ddx4^Cre/+* v.s. *Foxc2^f/+* mice). NOM, nominal; FDR, false discovery rate; ES, enrichment score; NES, normalized enrichment score.

The online version of this article includes the following source data and figure supplement(s) for figure 6:

**Source data 1.** Excel spreadsheet with the list of the differentially expressed genes found by single-cell RNA-sequencing (scRNA-seq) and enriched Gene Ontology terms.

**Figure supplement 1.** Single-cell RNA-sequencing (scRNA-seq) analysis of THY1⁺ undifferentiated spermatogonia (uSPG) in adult *Foxc2^f/+* and *Foxc2^f/-;Ddx4^Cre/+* mice.

**Figure supplement 2.** Re-cluster and developmental trajectory analysis of cells in Cluster1 derived from adult *Foxc2^f/+* and *Foxc2^f/-;Ddx4^Cre/+* mice.

---

SSCs, were isolated for CUT&Tag sequencing (*Figure 7A*). Specific peaks enriched in the promoter region of 3629 genes (*Figure 7B and C*; *Figure 7—source data 1*) showed functional enrichment in biological processes such as DNA repair and mitotic cell cycle regulation (*Figure 7D*). By overlapping with the 932 genes down-regulated in Cluster1 cells from *Foxc2^f/-;Ddx4^Cre/+* mice, we obtained 306 genes as the candidates subjective to the regulation by FOXC2 (*Figure 7E*; *Figure 7—source data 1*). Further, Gene Ontology (GO) enrichment analysis of these genes highlighted a distinctive functional cluster (11 genes) focusing on the negative regulation of cell cycle (*Figure 7F*; *Figure 7—source data 1*; *Bahar et al., 2002*; *Bakker et al., 2004*; *Gaudet et al., 2011*; *Georgescu et al., 2014*; *Knudson*

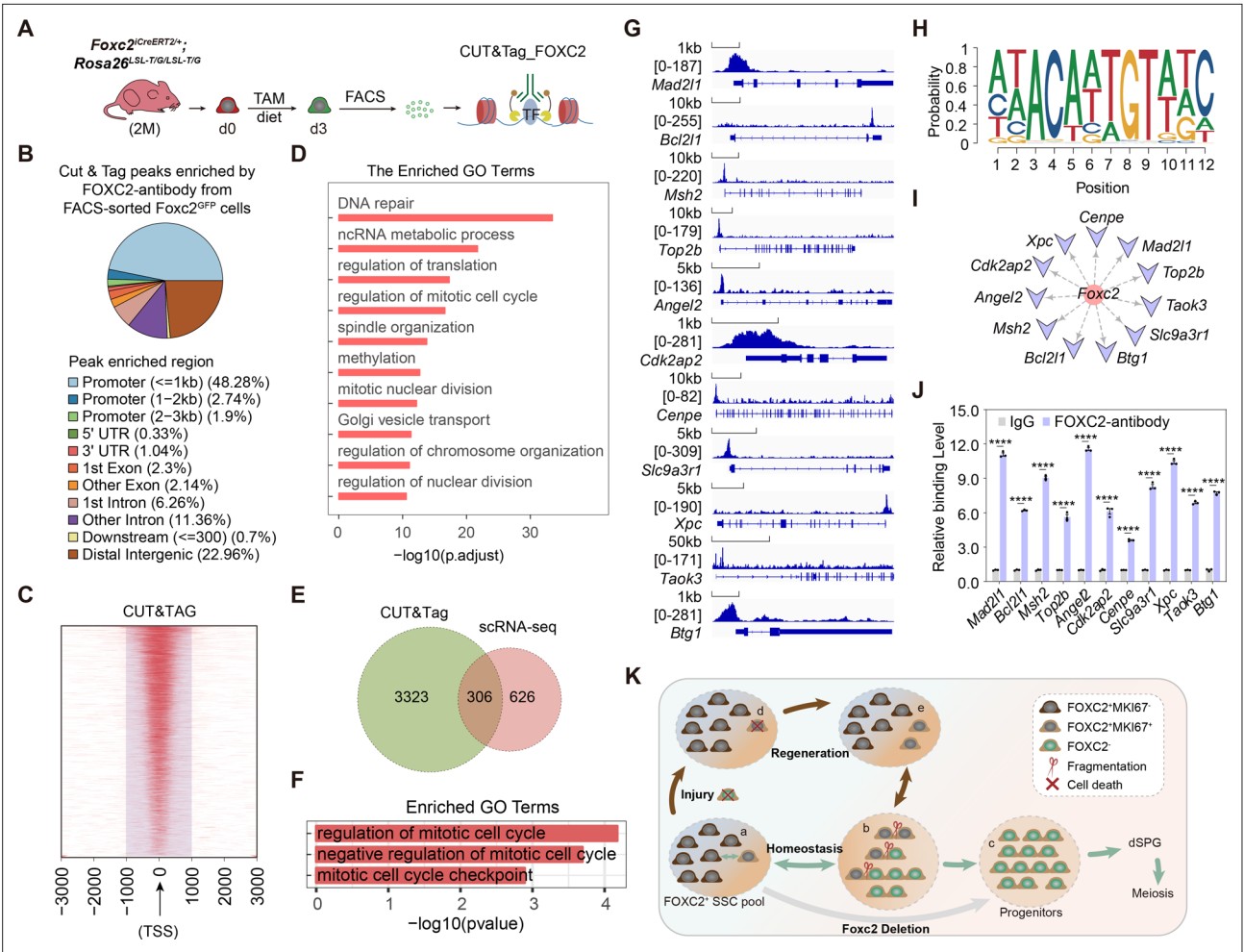

**Figure 7.** FOXC2 is essential for sustaining the quiescent state of spermatogonial stem cells (SSCs) via regulating cell cycle. (**A**) Workflow schematic illustration of the CUT&Tag_FOXC2 analysis on the fluorescence-activated cell sorting (FACS)-sorted FOXC2+ cells. (**B**) Pie chart for CUT&Tag_FOXC2 peaks genome distribution. (**C**) Profiling of CUT&Tag_FOXC2 peaks in proximity to transcriptional starting site (TSS). The distance to TSS within 1000 was highlighted in the purple box. (**D**) Top Gene Ontology (GO) terms enrichment by genes annotated by CUT&Tag_FOXC2 peaks. (**E**) Venn diagram of FOXC2 target genes defined by overlapping the CUT&Tag sequencing and single-cell RNA-sequencing (scRNA-seq) datasets. (**F**) GO terms enrichment by the FOXC2 target genes related to cell cycle regulation. (**G**) Chromatin landscapes of CUT&Tag_FOXC2 peaks of the candidates associated with negative cell cycle regulation. (**H**) The DNA-binding motif for FOXC2 (predicted with HOMER). (**I**) The cell cycle-related candidates possessing high binding potential (>0.8, predicted with JASPAR SCAN). (**J**) CUT&Tag-qPCR validation of the cell cycle arrest regulatory genes. (n=3). Values, mean ± s.e.m.; p-values were obtained using two-tailed t-tests (****p-value <0.0001). (**K**) The model for the maintenance of the FOXC2+ SSC subpopulation in adult testis.

The online version of this article includes the following source data for figure 7:

**Source data 1.** Excel spreadsheet with the list of the differentially expressed genes found by Cleavage Under Targets and Tagmentation (CUT&Tag) sequencing and Gene Ontology terms of the 306 crossed candidates.

and Korsmeyer, 1997; Nitiss, 2009; Raman et al., 2007; van Oosten et al., 2005; Weaver et al., 2003; Yi et al., 2012). More specifically, significant peaks enrichment at the promoter region were observed for these candidate genes (**Figure 7G**). Meanwhile, as predicted using the JASPAR Scan function (binding potential >0.8), there showed strong binding potential of FOXC2 toward these candidate genes (**Figure 7I**) via the binding motif of FOXC2 (**Figure 7H**), which was further confirmed by the result from the CUT&Tag qPCR (**Figure 7J**). Overall results imply that FOXC2 may function as a gatekeeper that ensures the quiescent state of SSCs by impeding cell cycle progression.

## Discussion

In this work, a comprehensive analysis of uSPG populations with scRNA-seq and the following lineage tracing study by whole-mount immunofluorescence assay led to the identification of FOXC2[+] SSCs as a quiescent SSC subpopulation in adult mice. Further investigation through functionality analysis confirmed FOXC2 is essential for SSC self-renewal and stemness, thereby is required for maintaining the SSC population that is critical for continuous spermatogenesis. Importantly, our data demonstrated that the colonies formed by FOXC2[+] cells constituted nearly the total length of the seminiferous tubules (99.31%), implying that the FOXC2[+] SSCs can support the complete spermatogenesis in adult mice.

GFRA1[+] $A_{pr}$ and $A_{al}$ cells are found to break randomly and a portion of them can return to the stem cell state (*Hara et al., 2014*). Interestingly, our findings showed that FOXC2 appeared in one of the $A_{pr}$ or $A_{al}$ cells at times, therefore raising a possibility that the subset of GFRA1[+] cells that return to stem cell state after intercellular bridge break, maybe FOXC2[+] due to different cell cycle state. If so, based on both findings, GFRA1[+]FOXC2[+] could represent a quiescent state whereas GFRA1[+]FOXC2[-] is proliferate active, which certainly requires further validation possibly through multiple lineage tracing and live imaging.

We observed that the FOXC2[+] SSCs were almost all in a non-proliferative state (~94.9%), and further revealed that FOXC2 functioned in the negative regulation of cell cycle progression, thus confirming that FOXC2-expressing SSCs are quiescent SSC population in adult mice. The finding that FOXC2 inhibited cell cycle and differentiation of SSCs in testis is consistent with that reported in other tissues (*Davis et al., 2004*; *Sabine et al., 2015*). In general, the quiescent state is a protective mechanism for stem cell storage and prevents stem cells from damage or depletion under genotoxic stresses (*Arai and Suda, 2007*; *Relaix and Zammit, 2012*; *Rodgers et al., 2014*; *Sharma et al., 2019b*). In our work, after the busulfan treatment, the quantity of FOXC2[+] cells remained stable and the survived uSPGs were predominantly FOXC2[+], indicating its insensitivity to cytotoxic agents. However, the proportion of MKI67[+]FOXC2[+] cells increased by 15.92% after 30 days of the busulfan treatment and decreased back to the pre-treatment level (5.08%) at 120 days, implying that the quiescent FOXC2[+] cells are able to transform into the proliferative FOXC2[+] cells to replenish the SSC pool to maintain the SSC homeostasis and normal spermatogenesis. We further confirmed by lineage tracing analysis that FOXC2-expressing cells were the only remaining SSC population and were responsible for germline regeneration after the busulfan treatment, indicating that FOXC2[+] SSCs represent a functionally important stem cell population with regenerative ability. In the future, more insights into the unique regulation of SSCs can be drawn from studying and comparing the transition between the quiescent and proliferative states in FOXC2[+] and other SSC subpopulations.

According to our findings, we propose a model for the maintenance of the FOXC2[+] SSC subpopulation (*Figure 7K*). Under physiological conditions, FOXC2[+] $A_s$ cells (including FOXC2[+]GFRA1[+], FOXC2[+]EOMES[+] cells, etc.) constitute the SSC pool, of which only a small proportion (~5.1%) cells are proliferative while the majority remains quiescent (*Figure 7Ka*). This population can divide symmetrically or asymmetrically into different $A_{pr}$ and $A_{al}$ (*Figure 7Kb*). Then FOXC2[+] cells (*Figure 7Kb*) may break from the syncytial and return to $A_s$ state (*Figure 7Ka*) to maintain the stable number of the SSC pool. FOXC2[-] progenies, derived from the FOXC2[+] population, form progenitors (*Figure 7Kc*) to support spermatogenesis. However, it requires continuous supply from the FOXC2[+] population and is subject to exhaustion when the supply is disrupted. In the context of regeneration conditions, the FOXC2[+]MKI67[-] cells can survive and set out the recovery process (*Figure 7Kd*). At the early stage, increasing proportion of FOXC2[+]MKI67[-] cells is committed to transforming into proliferative FOXC2[+]MKI67[+] cells, strengthening the supply to the progenitors (*Figure 7Ke*). At the late recovery stage, MKI67[+]/MKI67[-] ratio returns to the physiological level in FOXC2[+] population (*Figure 7Ka*), leaving the total number of FOXC2[+] cells stable, therefore maintaining the SSC homeostasis. However, it is necessary to perform more investigation to further improve and modify this model to gain a complete understanding of the connections between different SSC subpopulations in the testes of adult mice.

Based on our observation, FOXC2 seems nonessential for the transformation from gonocytes to SSCs in infant mice, in contrast to its requirement for adult spermatogenesis. A recent study showed that FOXC2[+] subpopulation in the postnatal mouse testis (<5 weeks) appeared more active in proliferation than the adult counterpart (*Wei et al., 2018*). Such differential functionality might reflect the

difference in the physical nature of spermatogenesis between developmental stages. For example, the maturity of spermatogenesis is still under development during the juvenile period with a focus on establishing and expanding the SSC pool. Therefore, it would be interesting to explore differences in individual functional contexts as well as the underlying regulatory mechanisms. Meanwhile, FOXC2, highly conserved between mice and humans with 94% identity in amino acid sequence (*Miura et al., 1997*), is also expressed in a subset of human adult SSCs, raising the possibility of an evolutionarily conserved mechanism governing SSC homeostasis in humans. Further work following this direction might be of great clinical significance specifically to patients who suffer from infertility. Moreover, the developmental correlation between FOXC2+ SSCs and other SSC subpopulations proposed previously should be revealed via biological methods such as multiple lineage tracing and live imaging. Collectively, our work here provides new insights into the investigation of adult SSCs and serves as a reference for studying the homeostasis and regeneration of other stem-cell systems.

# Materials and methods

## Mice

Animal experiments were approved by the Committee on Animal Care of the Institute of Basic Medical Sciences, Chinese Academy of Medical Sciences and Peking Union Medical College. The tab of animal experimental ethical inspection (No: ACUC-A01-2018-017) has been provided in supplementary file. The 8-week-old C57BL/6J wild-type mice were used for magnetic-activated cell sorting (MACS). The $Rosa26^{LSL-T/G/LSL-T/G}$ mice (stock no. 007676), $Ddx4^{Cre/+}$ mice (stock no. 000692) and EGFP$^{Tg/+}$ mice (stock no. 021930) were bought from the Jackson Laboratory. The $Foxc2^{iCreERt2}$ mice and the $Foxc2^{flox/flox}$ ($Foxc2^{f/f}$) mice were constructed and bought from the Biocytogen. The $Rosa26^{LSL-DTA/+}$ mice were bought from GemPharmatech. All mice were housed and bred under specific pathogen-free conditions (temperature: 22–26°C, humidity: 40–55%, 12 hr light/dark cycle) in the animal facility at the Institute of Basic Medical Sciences. DNA was isolated from the tails, and the genotypes of the mice were checked using PCR with specific primers (*Supplementary file 1*). All mice were randomly assigned to experiments and no statistical methods were used to predetermine sample size. The person performing the experiments did not know the sample identity until after data analysis. No data were excluded from analyses and the data displayed included a minimum of three independent experiments and a minimum of three biological replicates for each independent experiment. The 8-week-old C57BL/6J WT mice were treated with busulfan (40 mg/kg) and used as recipient mice 1 month later.

## Magnetic-activated cell sorting

The testes from 8-week-old C57BL/6J wild-type mice or 4-month-old $Foxc2^{f/+}$ and $Foxc2^{f/-};Ddx4^{Cre/+}$ mice (n=4) were minced and digested in the collagenase type IV (1 mg/mL, Sigma) and DNase I (500 µg/mL, Sigma) at 37°C for 15 min. The cell suspension was pipetted up and down once every 5 min and the digestion process was stopped with DMEM (containing 10% FBS). The cell suspension was filtered through a 40 µm nylon mesh, and after centrifugation, the cells were resuspended in 8 mL PBS. The 15 mL conical centrifuge tubes were slowly overlayed with 2 mL of 70% Percoll solution, 2 mL of 30% Percoll solution, and then 2 mL of testicular cell suspension and centrifuge at 600×*g* for 10 min at 4°C without using the centrifuge brake. After centrifugation, the cells at the interface between the 70% and the 30% Percoll solution were carefully removed into the new conical centrifuge tubes, washed with PBS, and then centrifuge at 600×*g* for 10 min at 4°C. After centrifugation, the cells were resuspended in 360 µL MACS buffer, added with 40 µL of magnetic microbeads conjugated with anti-Thy-1 antibody (Miltenyi Biotec 130049-101, Auburn, CA, USA), and mixed well. Incubate the cell suspension containing Thy-1 microbeads for 20 min at 4°C. Mix gently by tapping every 10 min. Add 20 mL of MACS buffer to the tube to dilute Thy-1 microbeads and centrifuge at 300×*g* for 10 min at 4°C. Remove the supernatant completely and resuspend in 2 mL of MACS buffer. Place the separation columns (MS Column; Miltenyi Biotec 130-042-201) in the magnetic field of the mini MACS Separation Unit (Miltenyi Biotec 130-142-102) and rinse with 0.5 mL of MACS buffer. Apply the cell suspension to the columns (500 µL/column). After the cell suspension has passed through the column and the column reservoir is empty, wash the column with 0.5 mL of MACS buffer three times. Remove the column from the MACS Separation Unit and elute the magnetically retained cells slowly into a 50 mL

conical centrifuge tube with 1 mL of MACS buffer using the plunger supplied with the column. Centrifuge the tube containing the cells at 600×$g$ for 10 min at 4°C and resuspend the cell pellet with 10 mL of MACS buffer for rinsing. Repeat this step once. After the final rinsing step, resuspend cells in 0.04% BSA and count the cell number.

## Single-cell RNA-seq

The MACS-sorted THY1[+] cells were used for loading onto the Chromium Single Cell 3' Chip kit v2 (10x Genomics, PN-120236) according to the instructions. Cell capturing and library preparation was performed following the kit instructions of the Chromium Single Cell 3' v2 Library and Gel Bead Kit (10x Genomics, PN-120237). In brief, 5000 cells were targeted for capture, and after cDNA synthesis, 10–12 cycles were used for library amplification. The libraries were then size-selected, pooled, and sequenced on a Novaseq 6000 (Illumina). Shallow sequencing was performed to access the library quality and to adjust the subsequent sequencing depth based on the capture rate and the detected unique molecular indices (UMI).

## scRNA-seq data processing

Raw sequencing reads were processed using the Cell Ranger v.3.0.1 pipeline of the 10x Genomics platform. In brief, reads from each sample were demultiplexed and aligned to the mouse mm10 genome, and UMI counts were quantified for each gene per cell to generate a gene-barcode matrix. Default parameters were used. The UMI counts were analyzed using the Seurat R Package (*Stuart et al., 2019*) (v.3.0.1) following the Seurat pipeline. Cells with more than 200 detected genes or less than 10% mitochondria reads were retained. Genes not detected in at least 10 cells were removed from subsequent analysis. The resulting matrix was normalized, and the most variable genes were found using Seurat's default settings, then the matrix was scaled with regression against the mitochondria reads. The top 2000 variable genes were used to perform PCA, and Jackstraw was performed using Seurat's default settings. Variation in the cells was visualized by UMAP for the top principal components. Cell types were determined using marker genes identified from the literature (*Kowalczyk et al., 2015*). We used the Seurat function CellCycleScoring to determine the cell cycle phase, as this program determines the relative expression of a large set of G2-M and S-phase genes. After removing the undefined cells, the spermatogonia were used for trajectory analysis, and the single-cell pseudotime trajectory was constructed with the Monocle 2 package (v2.12.0) (*Qiu et al., 2017a*; *Qiu et al., 2017b*; *Trapnell et al., 2014*) according to the provided documentation. The Monocle function clusterCells was used to detect cell clusters between clusters. The Seurat function FindAllMarkers with default settings was used to find DEGs upregulated in each cluster compared to the other cells. The top 200 DEGs of cluster1 were used for ordering cells, and the discriminative dimensionality reduction with trees (DDRTree) method was used to reduce the data to two dimensions. The dynamic expression patterns with the spermatogonial developmental trajectory of specific genes were visualized using the Monocle function plot_genes_in_pseudotime and plot_pseudotime_heatmap. The procession data of the adult human single-cell dataset was downloaded from Gene Expression Omnibus (GEO): GSE112013 (*Guo et al., 2018*) and the UMI counts were analyzed using the Seurat R Package (v.3.0.1) following the Seurat pipeline with the same parameters and functions as mentioned previously. According to the known markers, the germ cells characterized was used for trajectory analysis, and the single-cell pseudotime trajectory was constructed with the Monocle 2 package (v2.12.0) as mentioned previously.

## CUT&Tag sequencing and analysis

CUT&Tag assay was performed using CUT&Tag 2.0 High-Sensitivity Kit (Novoprotein scientific Inc, Cat# N259-YH01). The detailed procedures were described in *Kaya-Okur et al., 2019*; *Wang et al., 2021*. In brief, cells were harvested by trypsin and enriched by ConA-magnetic beads. 10,000 cells were resuspended in 100 mL Dig-wash Buffer (20 mM HEPES pH 7.5; 150 mM NaCl; 0.5 mM spermidine; 13 protease inhibitor cocktail; 0.05% digitonin) containing 2 mM EDTA and a 1:100 dilution of primary FOXC2 antibody. The primary antibody was incubated overnight at 4°C. Beads were washed in Dig-wash Buffer three times and incubated with secondary antibody for 1 hr at a dilution of 1:200. After incubation, the beads were washed three times in Dig-Hisalt Buffer (0.05% digitonin, 20 mM HEPES, pH 7.5, 300 mM NaCl, 0.5 mM spermidine, 13 protease inhibitor cocktail). Cells were

incubated with proteinA-Tn5 transposome at 25°C for 1 hr and washed three times in Dig-Hisalt buffer to remove unbound proteinA-Tn5. Next, cells were resuspended in 100 mL Tagmentation buffer (10 mM MgCl$_2$ in Dig-Hisalt Buffer) and incubated at 37°C for 1 hr. The tagmentation was terminated by adding 2.25 mL of 0.5 M EDTA, 2.75 mL of 10% SDS, and 0.5 mL of 20 mg/mL proteinase K at 55°C for 1 hr. The DNA fragments were extracted by phenol chloroform and used for sequencing on an Illumina HiSeq instrument (Illumina NovaSeq 6000) to generate 2×150 bp paired-end reads following the manufacturer's instructions.

Raw reads were analyzed by removing low-quality or adaptor sequences using Trim_galore (v0.5.0) and cleaned reads were mapped to the reference genome mm10 using Bowtie2 (v2.2.5). We used MACS2 (v2.1.2) to call peaks found in different groups. Homer (v4.11.1) de novo motif discovery tool was used for finding the binding motifs of Foxc2 with the findMotifsGenome.pl command. The binding potential of candidate target genes at the binding motif was predicated using the JASPAR Scan function (binding potential >0.8). The peaks filtered by fold change more than 5 and transcription start site less than 3000 bp were annotated by R package Chip Seeker for gene category analysis. R package Cluster profiler was used for gene function annotation such as KEGG and GO analysis.

## Enrichment analyses

GO and KEGG pathway enrichment analyses were conducted using the ClusterProfiler package (v3.12.0) (*Yu et al., 2012*) and the ClueGO app (v2.5.7) in Cytoscape (v3.8.1) with default settings and a p-value cut-off of 0.05. GSEA was assessed using the GSEA (v4.0.2) algorithm with MSigDB (v7.0) with default settings. The signaling pathways enriched by niche-derived paracrine factors and undifferentiated SPG-derived membrane proteins in the DEGs of the four samples were characterized. Then for each niche cell type, the niche-derived signaling pathways in all four samples were crossed with the SSC-derived signaling pathways to identify the candidate signaling pathways pivotal to SSCs maintenance.

## Transplantation assay

The 8-week-old C57BL/6J WT mice were treated with busulfan (40 mg/kg) and used as recipient mice 1 month later. SSCs were transplanted into the testis of recipient mice (1×10$^3$ cells/testis), and 2 months after transplantation, the testes were harvested and observed under a fluorescence microscope.

## Fluorescence-activated cell sorting

Single-cell suspensions were generated from testes or in vitro cultured SSCs. FACS was performed using an SH800 machine (Sony Biotechnology) to isolate the GFP$^+$ cells. Briefly, the GFP$^+$ gating area was based on the point of the fluorescence intensity axis where cells were considered as being GFP$^+$, set based on the background fluorescence intensity of a non-transgenic control testis cell population.

## Immunofluorescence

Mouse testes were fixed in 4% paraformaldehyde (PFA) at 4°C overnight, dehydrated, embedded in paraffin, and cut into 5 µm thick sections. The rehydrated mouse or human testis sections were subjected to antigen retrieval, blocked in 5% BSA with 0.1% Triton X-100, and incubated with primary antibody (*Supplementary file 1*) at 4°C overnight, including the germ cell marker DDX4, uSPG markers ZBTB16, LIN28A, ECAD (*Tokuda et al., 2007*), GFRA1, EOMES, PAX7, progenitor marker NEUROG3, and spermatocyte marker SYCP3 (*Yuan et al., 2000*). After three 5 min washes in PBS, the sections were incubated with secondary antibodies (*Supplementary file 1*) and DAPI (Sigma) at 37°C for 1 hr. After three 5 min washes in PBS, coverslips were then mounted on glass slides using anti-quencher fluorescence decay (Solarbio). Images were captured using a Zeiss 780 laser-scanning confocal microscope. Whole-mount immunofluorescence of seminiferous tubules was performed as previously described (*Di Persio et al., 2017*). Briefly, seminiferous tubules were disentangled from testicular biopsies and immediately fixed in 4% PFA at 4°C for 12 hr. After fixation, the seminiferous tubules were permeabilized with 0.5% Triton X-100 in PBS and treated with 5% BSA in PBS overnight at 4°C. After three 30 min washes, the seminiferous tubules were incubated with primary antibody (*Supplementary file 1*) overnight at 4°C. After three 30 min washes, the seminiferous tubules were incubated with species-specific secondary antibodies and DAPI at 4°C for 12 hr. After three 30 min

washes, the seminiferous tubules were mounted on slides with anti-quencher fluorescence decay (Solarbio) and observed with a Zeiss 780 laser-scanning confocal microscope.

## RNA isolation and quantitative RT-PCR analysis

Total RNA was extracted from the testes or cultured cells using the RNeasy kit (QIAGEN), reverse-transcribed using RevertAid First Strand cDNA Synthesis kit (Thermo), and processed for qRT-PCR using PowerUp SYBR Green Master Mix (Applied Biosystems) and a LightCycler 480 system (Roche) with gene-specific primers (*Supplementary file 1*). Reactions were run in triplicate and the mRNA levels were normalized to Gapdh and quantified using the delta-delta Ct method. The values shown are mean ± s.e.m. from three biological replicates.

## Tamoxifen inducible

According to a previous report for activation of iCre (*Sharma et al., 2019a*; *Sharma et al., 2019b*), the mice were fed with TD.130859 (TAM diet) for 3 days. The food was formulated for 400 mg tamoxifen citrate per kg diet, which would provide ~40 mg tamoxifen per kg body weight per day.

## Analyses of cyst length

The cyst length was obtained according to the previous report (*Nakagawa et al., 2010*). Briefly, to determine the cyst length, after immunofluorescence staining with anti-E-CAD antibody, the whole-mount seminiferous tubule specimens were observed under a fluorescence microscope. The E-CAD staining coupled with staining for FOXC2 enabled us to reliably identify syncytial cysts of FOXC2[+] cells.

## Analyses of cell density

The cell density was counted according to a previous report (*Tegelenbosch and de Rooij, 1993*). Briefly, the densities of the ZBTB16[+], GFRA1[+], LIN28A[+], or FOXC2[+] cells were measured on the seminiferous tubules with whole-mount staining, the numbers of which per 1000 Sertoli cells were determined.

## Sperm counts

Total sperm counts were obtained according to the previous report (*Roy et al., 2007*). Briefly, epididymal caput and cauda were minced and incubated in prewarmed M16 medium (Sigma-Aldrich) at 37°C in air containing 5% $CO_2$ for 30 min to allow the sperm to swim out. Then, the sperm were diluted in water and counted using a hemocytometer.

## Histology, evaluation of degenerating tubules

Testes of WT and mutant mice were fixed with PFA fixative and processed for paraffin-embedded section preparation (5 μm thick) and hematoxylin and eosin staining, according to standard procedures. The percentage of degenerating seminiferous tubules was calculated based on the cross-sections of seminiferous tubules (n>200) that appeared on one transverse section for each testis. In normal (WT) mouse testes, four generations of germ cells, each synchronously progressing through spermatogenesis, form cellular associations of fixed composition (called seminiferous epithelial stages). In the testes of *Foxc2*[f/-];*Ddx4*[Cre/+] mice, a few tubule cross-sections lacked one or more out of the four germ cell layers, which was defined as 'degenerative tubules' in this study.

## Statistical analysis

All statistical analyses were performed using GraphPad Prism (v7.0). All experiments were repeated at least three times, and data for evaluated parameters are reported as mean ± s.e.m. The p-values were obtained using two-tailed unpaired Student's t-tests or one-way ANOVA followed by Tukey test (ns represents p-value >0.05, * represents p-value <0.05, ** represents p-value <0.01, *** represents p-value <0.001, and **** represents p-value <0.0001).

## Acknowledgements

This work was supported by the National Key Research and Development Program of China grant (2022YFA0806302, 2018YFC1003500, 2019YFA0801800), CAMS Innovation Fund for Medical Sciences (CIFMS, 2021-I2M-1-019, 2017-I2M-3-009), National Natural Science Foundation of China grant (92268111, 31970794, 92268205, 32000586, 31725013, 32200646), and State Key Laboratory Special Fund grant (2060204).

## Additional information

### Funding

| Funder | Grant reference number | Author |
| --- | --- | --- |
| National Natural Science Foundation of China | 32200646 | Zhipeng Wang |
| National Natural Science Foundation of China | 31725013 | Jia Yu |
| National Natural Science Foundation of China | 32000586 | Kai Li |
| National Natural Science Foundation of China | 31970794 | Wei Song |
| National Natural Science Foundation of China | 92268111 | Wei Song |
| National Key Research and Development Program of China | 2022YFA0806302 | Wei Song |
| National Key Research and Development Program of China | 2018YFC1003500 | Wei Song |
| National Key Research and Development Program of China | 2019YFA0801800 | Jia Yu |
| National Natural Science Foundation of China | 92268205 | Jia Yu |
| CAMS Innovation Fund for Medical Sciences (CIFMS) | 2021-I2M-1-019 | Wei Song |
| CAMS Innovation Fund for Medical Sciences (CIFMS) | 2017-I2M-3-009 | Wei Song |
| State Key Laboratory Special Fund | 2060204 | Wei Song |

The funders had no role in study design, data collection and interpretation, or the decision to submit the work for publication.

### Author contributions

Zhipeng Wang, Conceptualization, Resources, Data curation, Formal analysis, Funding acquisition, Validation, Investigation, Visualization, Methodology, Writing – original draft, Project administration, Writing – review and editing; Cheng Jin, Conceptualization, Resources, Data curation, Software, Formal analysis, Validation, Investigation, Visualization, Methodology, Writing – original draft, Project administration, Writing – review and editing; Pengyu Li, Jielin Tang, Zhixin Yu, Jinhuan Ou, Dingfeng Zou, Mengzhen Li, Xinyu Mang, Jun Liu, Kai Li, Validation; Yiran Li, Software; Tao Jiao, Validation, Methodology; Han Wang, Software, Methodology; Yan Lu, Methodology; Ning Zhang, Shiying Miao, Resources, Supervision, Writing – review and editing; Jia Yu, Linfang Wang, Conceptualization, Resources, Supervision, Writing – review and editing; Wei Song, Conceptualization, Resources, Supervision, Writing – review and editing, Funding acquisition, Project administration, Writing – original draft, Data curation

### Author ORCIDs
Zhipeng Wang ⬤ http://orcid.org/0000-0001-9974-1703
Cheng Jin ⬤ http://orcid.org/0000-0002-3195-0653
Wei Song ⬤ http://orcid.org/0000-0002-8395-9991

### Ethics

Animal experiments were approved by the Committee on Animal Care of the Institute of Basic Medical Sciences, Chinese Academy of Medical Sciences and Peking Union Medical College. The tab of animal experimental ethical inspection (No: ACUC-A01-2018-017) has been provided in supplementary file.

Reviewer #1 (Public Review): https://doi.org/10.7554/eLife.85380.3.sa1
Reviewer #2 (Public Review): https://doi.org/10.7554/eLife.85380.3.sa2
Reviewer #3 (Public Review): https://doi.org/10.7554/eLife.85380.3.sa3
Author Response https://doi.org/10.7554/eLife.85380.3.sa4

## Additional files

### Supplementary files
- MDAR checklist
- Supplementary file 1. Excel spreadsheet with the primers and antibodies used in this study.

### Data availability

All data are available in the main text or supplementary materials. The scRNA-seq and CUT&Tag sequencing data produced in this study have been uploaded to the GEO (https://www.ncbi.nlm.nih.gov/geo) with accession codes GSE183163, GSE180729, and GSE180926. The expression of FOXC2 in adult human testis used the scRNA-seq dataset GSE112013 from previously published "The Human Testis Cell Atlas via Single-cell RNA-seq" by Jingtao Guo (*Guo et al., 2018*) (https://www.ncbi.nlm.nih.gov/geo/query/acc.cgi?acc=GSE112013). All of the R packages were available online and the code was used according to respective R packages documentation as described in the Methods. The MSigDB (v.7.0) used in this study is available at https://www.gsea-msigdb.org/gsea/msigdb.

The following datasets were generated:

| Author(s) | Year | Dataset title | Dataset URL | Database and Identifier |
| --- | --- | --- | --- | --- |
| Jin C, Wang Z | 2023 | 10X genomics scRNA-seq of testicular THY1+ cells in adult C57 mice | https://www.ncbi.nlm.nih.gov/geo/query/acc.cgi?acc=GSE183163 | NCBI Gene Expression Omnibus, GSE183163 |
| Jin C, Wang Z | 2023 | 10X genomics single-cell RNA sequencing of the testicular Thy1+ spermatogonia in Foxc2f/+ and Foxc2f/-;Ddx4-cre mice | https://www.ncbi.nlm.nih.gov/geo/query/acc.cgi?acc=GSE180729 | NCBI Gene Expression Omnibus, GSE180729 |
| Jin C, Wang Z | 2023 | CUT&Tag_Foxc2 for Foxc2+ cells from adult mice testes | https://www.ncbi.nlm.nih.gov/geo/query/acc.cgi?acc=GSE180926 | NCBI Gene Expression Omnibus, GSE180926 |

The following previously published dataset was used:

| Author(s) | Year | Dataset title | Dataset URL | Database and Identifier |
| --- | --- | --- | --- | --- |
| Guo J, Cairns BR | 2018 | The Human Testis Cell Atlas via Single-cell RNA-seq (Healthy men scRNA-seq data set) | https://www.ncbi.nlm.nih.gov/geo/query/acc.cgi?acc=GSE112013 | NCBI Gene Expression Omnibus, GSE112013 |

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
