## [Editor Report · eLife assessment]

This **important** study reports that Foxc2^+^ cells in the testis represent the quiescent spermatogonial stem cells (SSCs). The data supporting this claim are **solid**. The finding is of great significance to reproductive and stem-cell biology as male fertility depends on the fine balance between self-renewal and differentiation activities of the male germline stem cells, i.e., SSCs.

---

## [Referee Report · Reviewer #1 (Public Review)]

The expression and localization of Foxc2 strongly suggest that its role is mainly confined to As undifferentiated spermatogonia (uSPGs). Lineage tracing demonstrated that all germ cells were derived from the FOXC2+ uSPGs. Specific ablation of the FOXC2+ uSPGs led to the depletion of all uSPG populations. Full spermatogenesis can be achieved through the transplantation of Foxc2+ uSPGs. Male germ cell-specific ablation of Foxc2 caused Sertoli-only testes in mice. CUT&Tag sequencing revealed that FOXC2 regulates the factors that inhibit the mitotic cell cycle, consistent with its potential role in maintaining a quiescent state in As spermatogonia. These data made the authors conclude that the FOXC2+ uSPG may be the true SSCs, essential for maintaining spermatogenesis. The conclusion is supported by the data presented.

---

## [Referee Report · Reviewer #2 (Public Review)]

The authors found FOXC2 is mainly expressed in As of mouse undifferentiated spermatogonia (uSPG). About 60% of As uSPG were FOXC2+ MKI67-, indicating that FOXC2 uSPG were quiescent. Similar spermatogonia (ZBTB16+ FOXC2+ MKI67-) were also found in human testis.

The lineage tracing experiment using Foxc2CRE/+;R26T/Gf/f mice demonstrated that all germ cells were derived from the FOXC2+ uSPG. Furthermore, specific ablation of the FOXC2+ uSPGs using Foxc2Cre/+;R26DTA/+ mice resulted in the depletion of all uSPG population. In the regenerative condition created by busulfan injection, all FOXC2+ uSPG survived and began to proliferate at around 30 days after busulfan injection. The survived FOXC2+ uSPGs generated all germ cells eventually. To examine the role of FOXC2 in the adult testis, spermatogenesis of Foxc2f/-;Ddx4-cre mice was analyzed. From a 2-month-old, the degenerative seminiferous tubules were increased and became Sertoli cell-only seminiferous tubules, indicating FOXC2 is required to maintain normal spermatogenesis in adult testes. To get insight into the role of FOXC2 in the uSPG, CUT&Tag sequencing was performed in sorted FOXC2+ uSPG from Foxc2CRE/+;R26T/Gf/f mice 3 days after TAM diet feeding. The results showed some unique biological processes, including negative regulation of the mitotic cell cycle, were enriched, suggesting the FOXC2 maintains a quiescent state in spermatogonia.

Lineage tracing experiments using transgenic mice of the TAM-inducing system was well-designed and demonstrated interesting results. Based on all data presented, the authors concluded that the FOXC2+ uSPG are primitive SSCs, an indispensable subpopulation to maintain adult spermatogenesis. The conclusion of the mouse study is supported by the data presented.

---

## [Referee Report · Reviewer #3 (Public Review)]

By popular single-cell RNA-seq, the authors identified FOXC2 as an undifferentiated spermatogonia-specific expressed gene. The FOXC2+-SSCs can sufficiently initiate and sustain spermatogenesis, the ablation of this subgroup results in the depletion of the uSPG pool. The authors provide further evidence to show that this gene is essential for SSCs maintenance by negatively regulating the cell cycle in adult mice, thus well-established FOXC2 as a key regulator of SSCs quiescent state.

The experiments are well-designed and conducted, the overall conclusions are convincing. This work will be of interest to stem cell and reproductive biologists.

---

## [Author Response]

The following is the authors’ response to the original reviews.

**Reviewer #1 (Public Review):**
The expression and localization of Foxc2 strongly suggest that its role is mainly confined to As undifferentiated spermatogonia (uSPGs). Lineage tracing demonstrated that all germ cells were derived from the FOXC2+ uSPGs. Specific ablation of the FOXC2+ uSPGs led to the depletion of all uSPG populations. Full spermatogenesis can be achieved through the transplantation of Foxc2+ uSPGs. Male germ cell-specific ablation of Foxc2 caused Sertoli-only testes in mice. CUT&Tag sequencing revealed that FOXC2 regulates the factors that inhibit the mitotic cell cycle, consistent with its potential role in maintaining a quiescent state in As spermatogonia. These data made the authors conclude that the FOXC2+ uSPG may be the true SSCs, essential for maintaining spermatogenesis. The conclusion is largely supported by the data presented, but two concerns should be addressed: (1) terminology used is confusing: primitive SSCs, primitive uSPGs, transit amplifying SSCs... (2) the GFP+ cells used for germ cell transplantation should be better controlled using THY1+ cells.

Thanks for your good comments. According to your suggestions, we have addressed your two concerns as follows:

1> Overall our work suggest that FOXC2+ SSCs are a subpopulation of SSCs in a quiescent state, thus we have replaced the term ‘primitive’ with ‘quiescent’ in the revised manuscript. In general, ‘transient amplifying SSCs’ is considered to be ‘progenitors’, thus we have replaced ‘transient amplifying SSCs’ with ‘progenitors’ in the revised manuscript.

2> The transplantation experiment was conducted using MACS-sorted THY1+, FACS sorted THY1+, and FACS-sorted GFP+ (FOXC2+) uSPGs simultaneously. To be consistent with the single-cell RNA-seq using the MACS-sorted THY1+ uSPGs, we only presented the results from MACS-sorted THY1+ and FACS-sorted GFP+ (FOXC2+) uSPGs in the previous manuscript. Following the reviewer’s suggestion, we have included the results derived from FACS sorted THY1+ uSPGs as the control. The overall conclusion is still fully supported by the more comprehensive dataset, i.e. FOXC2+ cells generated significant higher numbers of colonies than THY1+ cells after transplantation (Figure 2D, E).

**Reviewer #2 (Public Review):**
The authors found FOXC2 is mainly expressed in As of mouse undifferentiated spermatogonia (uSPG). About 60% of As uSPG were FOXC2+ MKI67-, indicating that FOXC2 uSPG were quiescent. Similar spermatogonia (ZBTB16+ FOXC2+ MKI67-) were also found in human testis.The lineage tracing experiment using Foxc2iCreERT2/+;Rosa26LSL-T/G/LSL-T/G mice demonstrated that all germ cells were derived from the FOXC2+ uSPG. Furthermore, specific ablation of the FOXC2+ uSPGs using Foxc2iCreERT2/+;Rosa26LSL-DTA/+ mice resulted in the depletion of all uSPG population. In the regenerative condition created by busulfan injection, all FOXC2+ uSPG survived and began to proliferate at around 30 days after busulfan injection. The survived FOXC2+ uSPGs generated all germ cells eventually. To examine the role of FOXC2 in the adult testis, spermatogenesis of Foxc2f/-;Ddx4Cre/+ mice was analyzed. From a 2-month-old, the degenerative seminiferous tubules were increased and became Sertoli cell-only seminiferous tubules, indicating FOXC2 is required to maintain normal spermatogenesis in adult testes. To get insight into the role of FOXC2 in the uSPG, CUT&Tag sequencing was performed in sorted FOXC2+ uSPG from Foxc2iCreERT2/+;Rosa26LSL-T/G/LSL-T/G mice 3 days after TAM diet feeding. The results showed some unique biological processes, including negative regulation of the mitotic cell cycle, were enriched, suggesting the FOXC2 maintains a quiescent state in spermatogonia.Lineage tracing experiments using transgenic mice of the TAM-inducing system was well-designed and demonstrated interesting results. Based on all data presented, the authors concluded that the FOXC2+ uSPG are primitive SSCs, an indispensable subpopulation to maintain adult spermatogenesis.The conclusion of the mouse study is mostly supported by the data presented, but to accept some of the authors' claims needs additional information and explanation. Several terminologies define cell populations used in the paper may mislead readers.1. "primitive spermatogonial stem cell (SSC)" is confusing. SSCs are considered the most immature subpopulation of uSPG. Thus, primitive uSPGs are likely SSCs. The naming, primitive SSCs, and transit-amplifying SSCs (Figure 7K) are weird. In general, the transit-amplifying cell is progenitor, not stem cell. In human and even mouse, there are several models for the classification of uSPG and SSCs, such as reserved stem cells and active stem cells. The area is highly controversial. The authors' definition of stem cells and progenitor cells should be clarified rigorously and should compare to existing models.

Thanks for your good comments. Considering that our results showed that FOXC2+ SSCs are in a quiescent state and that Mechanistically FOXC2 maintained the quiescent state of SSCs by promoting the expression of negative regulators of cell cycle, we have replaced ‘primitive SSCs’ with ‘quiescent SSCs’ in the revised manuscript. We agree with the reviewer that ‘transient amplifying SSCs’ is considered to be ‘progenitors’, thus we have replaced ‘transient amplifying SSCs’ with ‘progenitors’ in the revised manuscript. Further，from our point of view, the FOXC2+Ki67+ SSCs could be regarded as active stem cells, and the FOXC2+Ki67- SSCs could be regarded as reserved stem cells, although further research evidence is still needed to confirm this.

1. scRNA seq data analysis and an image of FOXC2+ ZBTB16+ MKI67- cells by fluorescent immunohistochemistry are not sufficient to conclude that they are human primitive SSCs as described in the Abstract. The identity of human SSCs is controversial. Although Adark spermatogonia are a candidate population of human SSCs, the molecular profile of the Adark spermatogonia seems to be heterogeneous. None of the molecular profiles was defined by a specific cell cycle phase. Thus, more rigorous analysis is required to demonstrate the identity of FOXC2+ ZBTB16+ MKI67- cells and Adark spermatogonia.

We agree with the reviewer that the identity of human SSCs remain elusive even though Adark population demonstrates certain characteristics of SSCs. To acknowledge this notion, we have revised our conclusion as such that only suggests FOXC2+ZBTB16+MKI67- represents a quiescent state of human SSCs.

1. FACS-sorted GFP+ cells and MACS-THY1 cells were used for functional transplantation assay to evaluate SSC activity. In general, the purity of MACS is significantly lower than that of FACS. Therefore, FACS-sorted THY1 cells must be used for the comparative analysis. As uSPGs in adult testes express THY1, the percentage of GFP+ cells in THY1+ cells determined by flow cytometry is important information to support the transplantation data.

Thanks for your good comments. According to your suggestions, we have addressed your concerns as follows:

1> The transplantation experiment was conducted using MACS-sorted THY1+, FACS sorted THY1+, and FACS-sorted GFP+ (FOXC2+) uSPGs simultaneously. To be consistent with the single-cell RNA-seq using the MACS-sorted THY1+ uSPGs, we only presented the results from MACS-sorted THY1+ and FACS-sorted GFP+ (FOXC2+) uSPGs in the previous manuscript. Following the reviewer’s suggestion, we have included the results derived from FACS sorted THY1+ uSPGs as the control. The overall conclusion is still fully supported by the more comprehensive dataset, i.e. FOXC2+ cells generated significant higher numbers of colonies than THY1+ cells after transplantation (Figure 2D, E).

2> We performed FACS analysis to determine the proportion of GFP+ cells in FACS-sorted THY1+ cells from Rosa26LSL-T/G/LSL-T/G or Foxc2iCreERT2/+;Rosa26LSL-T/G/LSL-T/G mice at day 3 post TAM induction, and the result showed that GFP+ cells account for approximately 20.9±0.21% of THY1+ cells, See Author response image 1.

**Author response image 1. sa4fig1:** 

1. The lineage tracing experiments of FOXC2+-SSCs in Foxc2iCreERT2/+;Rosa26LSL-T/G/LSL-T/G showed ~95% of spermatogenic cells and 100% progeny were derived from the FOXC2+ (GFP+) spermatogonia (Figure 2I, J) at month 4 post-TAM induction, although FOXC2+ uSPG were quiescent and a very small subpopulation (~ 60% of As, ~0.03% in all cells). This means that 40% of As spermatogonia and most of Apr/Aal spermatogonia, which were FOXC2 negative, did not contribute to spermatogenesis at all eventually. This is a striking result. There is a possibility that FOXC2CRE expresses more widely in the uSPG population although immunohistochemistry could not detect them.

Thanks for your good comments. From our lineage tracing results, over 95% of the spermatogenic cells are derived from the FOXC2+ SSCs in the testes of 4-month-old mice, which means that FOXC2+ SSCs maintain a long-term stable spermatogenesis. In addition, previous studies have shown that only a portion of As spermatogonia belong to SSCs with complete self-renewal ability (PMID: 28087628, PMID: 25133429), which is consistent with our findings. Therefore, we speculate that 40% of As spermatogonia and most of Apr/Aal spermatogonia, which were FOXC2 negative, did contribute to spermatogenesis but cannot maintain a long-term spermatogenesis due to limited self-renewal ability.

1. The CUT&Tag_FOXC2 analysis on the FACS-sorted FOXC2+ showed functional enrichment in biological processes such as DNA repair and mitotic cell cycle regulation (Figure 7D). The cells sorted were induced Cre recombinase expression by TAM diet and cut the tdTomato cassette out. DNA repair process and negative regulation of the mitotic cell cycle could be induced by the Cre/lox recombination process. The cells analyzed were not FOXC2+ uSPG in a normal physiological state.

We do appreciate the reviewer’s concern on the possibility of the functions enriched in the analysis as referred might be derived from Cre/lox recombination. However, we think it is unlikely that the Cre/lox recombination process, supposed to be rather local and specific, can trigger such a systemic and robust response by the DNA damage and cell cycle regulatory pathways. The reasons are as follows: First, as far as we are aware, there has been sufficient data to support this suggested scenario. Second, we did not observe any alteration in either the SSC behaviors or spermatogenesis in general upon the TAM-induced genomic changes, suggesting the impact from the Cre/lox recombination on DNA damage or cell cycle was not significant. Third, no factors associated with the DNA repair process were revealed in the differential analysis of single-cell transcriptomes of FOXC2-WT and FOXC2-KO.

1. Wei et al (Stem Cells Dev 27, 624-636) have published that FOXC2 is expressed predominately in As and Apr spermatogonia and requires self-renewal of mouse SSCs; however, the authors did not mention this study in Introduction, but referred shortly this at the end of Discussion. Their finding should be referred to and evaluated in advance in the Introduction.

Thanks for your good comments. According to your suggestion, we have revised the introduction to refer this latest parallel work on FOXC2. We are happy to see that our discoveries are converged to the important role of FOXC2 in regulating SSCs in adult mammals.

**Reviewer #3 (Public Review):**
By popular single-cell RNA-seq, the authors identified FOXC2 as an undifferentiated spermatogonia-specific expressed gene. The FOXC2+-SSCs can sufficiently initiate and sustain spermatogenesis, the ablation of this subgroup results in the depletion of the uSPG pool. The authors provide further evidence to show that this gene is essential for SSCs maintenance by negatively regulating the cell cycle in adult mice, thus well-established FOXC2 as a key regulator of SSCs quiescent state.The experiments are well-designed and conducted, the overall conclusions are convincing. This work will be of interest to stem cell and reproductive biologists.

Thanks for the positive feedback.

**Reviewer #1 (Recommendations for the Authors):**
The authors should address the following concerns:1. The most primitive uSPGs should be the true SSCs. The term "primitive SSCs" is very confusing.1. In addition to FACS-sorted GFP+ cells, FACS-sorted THY1+ cells should also be used for transplantation.

Thanks for your good comments. According to your suggestions, we have addressed your two concerns as follows:

1. Overall our work suggest that FOXC2+ SSCs are a subpopulation of SSCs in a quiescent state, thus we have replaced the term ‘primitive’ with ‘quiescent’ in the revised manuscript.

2. The transplantation experiment was conducted using MACS-sorted THY1+, FACS sorted THY1+, and FACS-sorted GFP+ (FOXC2+) uSPGs simultaneously. To be consistent with the single-cell RNA-seq using the MACS-sorted THY1+ uSPGs, we only presented the results from MACS-sorted THY1+ and FACS-sorted GFP+ (FOXC2+) uSPGs in the previous manuscript. Following the reviewer’s suggestion, we have included the results derived from FACS sorted THY1+ uSPGs as the control. The overall conclusion is still fully supported by the more comprehensive dataset, i.e. FOXC2+ cells generated significant higher numbers of colonies than THY1+ cells after transplantation (Figure 2D, E).

**Reviewer #3 (Recommendations for the Authors):**
The experiments are well-designed and conducted, the overall conclusions are convincing. The only concerns are the writing, especially the introduction which was not well-rationalized. Sounds the three subtypes and three models for SSCs' self-renew are irrelevant to the major points of this manuscript. I don't think you need to talk too much about the markers of SSCs. Instead, I suggest you provide more background about the quiescent or activation states of the SSCs. In addition to that, as a nuclear-localized protein, it cannot be used to flow cytometric sorting, I don't think it should be emphasized as a marker. You identified a key transcription factor for maintaining the quiescent state of the primitive SSCs, that's quite important!

Appreciate the positive feedback and constructive suggestions on the writing. We have substantially revised our manuscript to include the relevant advances and understanding from the field as well as highlight the importance of FOXC2 in regulating the quiescent state of SSCs.